# The Impacts of Woolly Cupgrass on the Antioxidative System and Growth of a Maize Hybrid

**DOI:** 10.3390/plants10050982

**Published:** 2021-05-14

**Authors:** Arnold Szilágyi, László Radócz, Mária Takácsné Hájos, Csaba Juhász, Béla Kovács, Gabriella Kovács, Erika Budayné Bódi, Csaba Radványi, Makoena Joyce Moloi, Lóránt Szőke

**Affiliations:** 1Institute of Plant Protection, University of Debrecen, 138 Böszörményi St., 4032 Debrecen, Hungary; szilagyi.arnold@agr.unideb.hu (A.S.); kovacs.gabriella@agr.unideb.hu (G.K.); radvanyic@gmail.com (C.R.); 2Institute of Horticultural Science, University of Debrecen, 138 Böszörményi St., 4032 Debrecen, Hungary; hajos@agr.unideb.hu; 3Arid Land Research Centre, University of Debrecen, 138 Böszörményi St., 4032 Debrecen, Hungary; juhasz@agr.unideb.hu (C.J.); bodi.erika@agr.unideb.hu (E.B.B.); 4Institute of Food Science, University of Debrecen, 138 Böszörményi St., 4032 Debrecen, Hungary; kovacsb@agr.unideb.hu (B.K.); szoke.lorant@agr.unideb.hu (L.S.); 5Department of Plant Sciences, University of the Free State-Main Campus, P.O. Box 339, Bloemfontein 9300, South Africa; MoloiMJ@ufs.ac.za

**Keywords:** antioxidant enzymes, woolly cupgrass, allelopathy, protein, morphological parameters

## Abstract

Woolly cupgrass (*Eriochloa villosa* (Thunb.) Kunth) is a new invasive weed in Hungary. This study was conducted to elucidate the effects of this weed on the biochemistry and growth of maize (*Zea mays* L. cv. Armagnac) under greenhouse conditions. Activities of the antioxidative enzymes (ascorbate peroxidase (APX), guaiacol peroxidase (POD), superoxide dismutase (SOD)), the contents of malondialdehyde (MDA), and protein were measured in the shoots and roots, whereas the content of the photosynthetic pigments was measured only in the shoots. The measured growth parameters included plant height, root length, root volume, root and shoot dry weight, and stem diameter. Results showed the allelopathic effects of woolly cupgrass on maize, with significant decreases in plant height, root length, root volume, and root dry weight. Woolly cupgrass infestation (WCI) induced significantly higher activities of APX and SOD in the shoots, whereas POD was only induced in the roots. The contents of chlorophyll-a, total chlorophyll (including relative chlorophyll), carotenoids, and root protein were substantially reduced by WCI, except for the leaf chlorophyll-b. The results suggest that high APX and SOD activities in the shoots could be involved in stabilizing the leaf chlorophyll-b, chlorophyll a/b, shoot protein, and shoot dry weight because all of these parameters were not inhibited when these two enzymes were induced. In contrast, high activity of POD in the roots is not effective in counteracting allelopathy. Therefore, it would be worthwhile to further investigate if an increase in the activities of APX and SOD in the shoots of WCI maize is responsible for stabilizing leaf chlorophyll-b, shoot protein, and shoot dry weight, which could contribute to improved maize yield under WCI.

## 1. Introduction

Invasive alien plant species negatively affect ecosystem structure and function at multiple tiers [1,2]. These species often disturb the composition of native species and reduce vegetative biodiversity by dominating the landscape of the invaded area. Introduction of new alien weeds in crops has become a major issue for farmers in Central Europe [3,4]. A substantial number of alien plants are found in crop fields, while others are considered to be at an early stage of invasion (“emerging weeds”) and are expected to become even more problematic in the future [4,5]. This may be accelerated by the changing climate and its accompanying environmental constraints, which may alter competitive interactions between agricultural weeds and cultivated crops [6], aiding the spread of invasive plants to new areas [7].

Woolly cupgrass (*Eriochloa villosa* (Thunb.) Kunth) is an invasive weed species with immense ecological and agricultural impacts [8]. It is an annual C_4_ grass of East Asian origin (China and bordering countries) [9] and is classified as a T_4_ group plant. It is a noxious weed in cultivated crops, lea-land, and sometimes in gardens. Because woolly cupgrass is well adapted to agro-ecosystems in arable land, it may be expected to be problematic in horticultural systems because of its relationships with *Panicum*, *Setaria*, and *Digitaria* species. Globally, weed-associated maize losses are estimated at 10.5% [10]. Woolly cupgrass causes significant losses in field crops, especially in the United States of America (USA) and Europe, resulting in increased cost of weed control [11]. The control of woolly cupgrass by herbicides is not effective. Herbicides are applied in May but germination of this weed is active from mid-April to the first frost in the USA and Europe, making herbicides ineffective against newly germinated woolly cupgrass [11,12,13,14]. In addition to maize, woolly cupgrass is a harmful weed of sunflower (*Helianthus annuus* L.) and soybean (*Glycine max* L., Merr.) especially in Russia, Czech Republic, Austria, Hungary, and Romania [15]. In the USA, research shows that it is present in 11 primary maize production areas (important areas: Iowa, Illinois, Minnesota, Wisconsin, etc.) [11,13,14]. In addition, it has been recorded in Canada [14], where it was added to the Weed Seeds Order in 2005 as a prohibited noxious weed [9]. 

Among other factors (light, nutrient, and moisture), weeds can also affect crop growth through allelopathy [16]. Allelopathy refers to the inhibition of plant growth by allelochemicals released by another plant species. Research advancements broadened the definition of allelopathy to include all direct positive or negative effects of one plant on another plant or microorganisms by the liberation of biochemicals into the natural environment [17]. Allelochemicals usually affect the plant roots (root length and mean root diameter). Furthermore, they influence the levels of reactive oxygen species (ROS) [18] such as singlet oxygen (_1_O^2^), superoxide anion (O^2−^), hydrogen peroxide (H_2_O_2_), and hydroxyl radical (OH^-^). These radicals are able to oxidize vital cellular ultrastructure and lead to oxidative damage, lipid peroxidation, and devastation of cellular organelles [19,20]. However, to counteract the unfavorable influences of the free radicals, plants may possess an antioxidative system (this is a non-enzymatic and enzymatic system) that reduces the effects of oxidative stress [21,22].

There is limited knowledge on the influences of allelochemicals released by woolly cupgrass on the biochemical and morphological aspects of maize. Therefore, this study investigated the influence of woolly cupgrass on the root and shoot lipid peroxidation, protein contents, activities of the antioxidative enzymes (ascorbate oxidase, guaiacol peroxidase, and superoxide dismutase), leaf photosynthetic pigments, and growth parameters (plant height, root length, root volume, root and shoot dry weight, and stem diameter) of a Hungarian maize hybrid (*Zea mays* L. cv. Armagnac). The results may identify valuable parameters that could assist in the future breeding for tolerance of woolly cupgrass infestation.

## 2. Results

The allelopathic effect of woolly cupgrass on the photosynthetic pigments of maize was observed. The relative-chlorophyll content of maize leaves was significantly decreased by woolly cupgrass at both sampling times (approximately 20%, 14 d.a.s. (days after sowing) and 19%, 21 d.a.s.) (Figure 1). Similarly, the allelopathic effect of woolly cupgrass was observed on the fifth leaf where the relative-chlorophyll content decreased by 18.20% (results not shown).

Because relative-chlorophyll content is a relative value, the quantity of individual photosynthetic pigments was also determined. The results are presented in Figure 2.

The negative effects of woolly cupgrass were observed on the chlorophyll-a (15% reduction)*,* carotenoids (20% reduction), and chlorophyll-a + b (17% reduction). The mean chlorophyll-b and chlorophyll-a/b contents were not affected by the allelopathic effects of woolly cupgrass.

Plant height was reduced by infestation of maize with woolly cupgrass 14 d.a.s. (25%) and 21 d.a.s. (22%) (Figure 3).

The WCI also affected the root parameters negatively by reducing the total root length, root volume, root surface, and the root diameter fractions (0.2, 0.4, 0.6, 0.8, and 1 mm). Although lower average diameter was observed due to the allelopathic effect of woolly cupgrass, this reduction was not significant (Table 1).

Woolly cupgrass had no significant allelopathic effect on the stem diameter at either sampling time. These results are not shown in the text.

The plant dry weights decreased due to the woolly cupgrass allelopathic effect. This reduction, however, was significant for the root + shoot and root dry weights compared to the control plants (56% and 31%, respectively). There was no significant shoot dry weight difference between the WCI and control plants (Figure 4).

The allelopathic effects of woolly cupgrass were also observed on some of the biochemical parameters. The WCI induced significantly high MDA (malondialdehyde) content in the shoot (23%) and roots (71%) compared to the control plants (Figure 5).

The WCI induced higher APX (28%) and SOD (16%) activities in the shoots compared to the control plants. In contrast, in the roots, there were no significant changes compared to the control plants (Figure 6 and Figure 7).

In contrast to the above-mentioned enzymes (APX, SOD), woolly cupgrass induced a substantial increase in POD activity in the roots. There were no significant differences in the POD activity of the control shoots and those of WCI maize (Figure 8).

The WCI significantly decreased the protein content in the roots. In the shoots however, the reduction was not significant (Figure 9). 

## 3. Discussion

The main goal of this study was to examine the effects of woolly cupgrass on maize, at morphological and biochemical levels under greenhouse conditions, because there are no records of such studies. Some weeds and host plants produce allelochemicals, which may impair each other’s performance [23].

Statistical analysis indicates that the contents of leaf relative chlorophyll (Figure 1), total chlorophyll, chlorophyll-a, and carotenoids (Figure 2) significantly decreased in the woolly cupgrass infested (WCI) maize compared to the controls. However, it appears that the allelopathic effect of woolly cupgrass is somehow selective because the chlorophyll-b content and chlorophyll-a/b ratio were not significantly reduced. This implies that photosynthesis could still take place; however, the capacity could be lower during WCI. Based on the results, it is clear that WCI does not degrade chlorophyll-b in maize. Therefore, factors leading to this need to be further studied, which will be valuable for future maize production under WCI. The allelopathic effects of the different plant species on chloroplast pigments have been reported. Farhoudi and Lee [24] stated that barley extract inhibited the chlorophyll-a, -b content of *Hordeum sponteneum* and *Avena ludovician* (a non-selective effect). In agreement, Farooq et al. [25] showed that the allelopathic effect of tobacco leads to the reduction of chlorophyll-a, -b content in maize and mung bean. Furthermore, the relative-chlorophyll content (SPAD unit) was inhibited by the allelopathic effect of spearmint (*Mentha spicata* L.) in maize [26]. In contrast, the chlorophyll content was not reduced by the allelopathic effect of *Tithonia diversifolia* in the maize leaf [27]. This suggests that the effect of the allelochemicals released is closely associated with the species type.

Substantial allelopathic effects of woolly cupgrass on maize height, root length, and volume were observed in this study (Figure 3 and Table 1). Future studies are needed to determine whether these reductions are influenced by reduced photosynthetic capacity. Different plant species can have allelopathic effects on others by affecting plant height, number shoots, and number of roots. For example, Abdul-Raoof and Siddiqui [28] stated that *Tinospora cordifolia* extract reduced the *Chenopodium* and *Cassia* spp. shoot and root length. In addition, *Datura stramonium* reduced *Cenchrus ciliaris* and *Neonotonia wightii* shoot and root length [29]. Similarly, Kabir et al. [30] observed the allelopathic effects of rice varieties on spinach root and shoot length. Motamedi et al. [31] further showed that *Carthamus tinctorius* decreased the height of *Amaranthus* spp. Despite the reduction of shoots and roots, stem diameter was not affected by allelochemicals from woolly cupgrass. In agreement, extracts from *Panicum ruderale* did not reduce shoot and root length significantly in maize, whereas *Aslcepias syriaca* and *Abutilon theoprasi* significantly reduced the above-mentioned parameters [32].

In addition, root dry weight was significantly reduced by woolly cupgrass (Figure 4), further confirming the allelopathic effect of this weed on maize. Several studies have also reported a similar effect [33,34]. In contrast, the shoot dry weight was not affected by WCI. Based on these results, it is important for future research to focus on whether the stability of shoot dry weight is related to the leaf photosynthetic capacity of WCI maize or not.

In this study, WCI substantially increased the MDA content (an equivalent of lipid peroxidation) in the shoots and roots (Figure 5), further supporting the allelopathic effects of this weed on maize because high lipid peroxidation in plants is an indicator of more cellular damage by oxidative burst. Similar increases in MDA were also recorded in wheat/rape–*Phytolacca latbenia* interaction [35]. The lipid peroxidation-associated allelopathic effects of *Mikania micrantha* and *Ipomoea cairica* (other invasive plants) have been documented in *Chrysanthemum coronarium* [36].

WCI led to significantly high APX and SOD activities in the maize shoots (Figure 6 and Figure 7). In the roots however, there were no changes in these enzyme activities when exposed to WCI, showing that their activities are specifically related to the plant organ. Hatata and El-Darier [37] found that the activity of APX was higher in wheat leaves when treated with *Achillea santolina* shoot extract. Gindri et al. [38] also stated that *Lantana camara* L. increased the APX activity of the host plant (*Avena sativa* L.) seedling. According to Ding et al. [39], higher SOD activity was measured in garlic (*Allium sativum* L.) root exudates treated with lettuce extract, which contradicts with our results. Kapoor et al. [40] stated that *Artemisia absinthium* and *Psidium guajava* extract increased the SOD activity of *Parthenium hysterophorus* leaves. Similarly, woolly cupgrass induced POD activity selectively in the roots (Figure 8). El-Shora and El-Gawad [41] reported that *Rumex dentatus* extract substantially increased the POD activity in *Cicer arietinum* root. In contrast, Madany and Saleh [42] discovered the allelopathic effect of *Euphorbia helioscopia* on the POD activity of wheat and pea seedlings. The role of antioxidant enzymes in plants, especially during stress, is to ensure survival of the plants by maintaining cellular homeostasis between production of the reactive oxygen species (ROS) and ROS scavenging [43]. Therefore, during woolly cupgrass–maize association, it appears that increases in the activities of the antioxidant enzymes (whether in the roots or shoots) were not effective in the prevention of lipid peroxidation. However, the role of induced shoot APX and SOD activities during WCI should be further studied because they corresponded to stable shoot dry weight and chlorophyll-b. In addition, it is important to undertake such a study to determine whether the enzymes are important for maize survival under WCI, because it was elsewhere reported that combined activities of APX and SOD are important for protection and survival of common bean cultivars [44].

The allelopathic effect of woolly cupgrass on the root protein content was observed (significant decrease) (Figure 9). Similar results were reported in wild barley (*Hordeum spontaneum* L.) and great brome (*Bromus diandrus* Roth., syn. *Bromus rigidus* Roth. *subsp. gussonii* Parl.) due to the allelopathic effects of *Achillea biebersteinii* and barley (*Hordeum vulgare* L. *subsp. vulgare*) [45,46]. Interestingly, in the shoot, no significant reduction in protein was recorded in the WCI maize compared to the control, which prompts the need to understand whether the induced activities of APX and SOD in the shoot could be associated with the observed shoot protein stability.

In conclusion, woolly cupgrass has allelopathic effects on maize because it reduced most of the maize morphological traits (plant height, root length, and root dry weight). Specific biochemical parameters associated with this allelopathy include reduced leaf total chlorophyll (attributed to low chlorophyll-a content), carotenoid, and root protein contents, and increased leaf and root lipid peroxidation. The activities of APX and SOD were selectively induced to high levels in the shoots of WCI maize, which coincided with no significant reduction in the leaf chlorophyll-b, chlorophyll a/b, shoot protein, and shoot dry weight. In contrast, increased POD activity in the roots coincided with reduced root protein and reduced root parameters. Therefore, it would be worthwhile to further investigate if the increases in the activities of APX and SOD in the shoots of WCI maize are responsible for stabilizing leaf chlorophyll-b, shoot protein, and shoot dry weight, which consequently can contribute to maize yield under WCI.

## 4. Materials and Methods

### 4.1. Plant Materials and Growth Conditions

The plants (*Zea mays* L. cv. Armagnac) were grown in the greenhouse of the Institute of Horticultural Science (University of Debrecen, Debrecen, Hungary). Humidity was 50–60% and the temperature was 34/26 °C day/night. Plants were planted (one plant/pot) in peat with the following composition: 70% brown and 30% medium ripe Sphagnum moss peat with the addition of 1.5 kg/m^3^ Multi mix fertilizer, 14–16–18% MeO Hand water-binding additive, when the woolly cupgrass was four-five leaf stage. The pot diameter was 11 cm, and the height was 24 cm. The control treatment was without woolly cupgrass in the pots. Irrigation to 60–70% pot field capacity was carried out daily with a volume of 0.5 L of water per pot. Five replications of plants per treatment were used for morphological (height, stem diameter, root parameters), biochemical (relative-chlorophyll contents, photosynthetic pigment, and the rate of lipid peroxidation and enzymology) parameters. There were two sampling times, 14 and 21 d.a.s., when the maize plants were at the 4 and 5 leaf stage respectively. The woolly cupgrass infested plants were labelled WCI in the text.

### 4.2. Quantification of the Photosynthetic Pigments

The relative-chlorophyll content was measured on the fourth leaf (14 d.a.s.) and fifth leaf (21 d.a.s.) on ten plants in five repetitions. The measurement was undertaken using a SPAD-502 Plus chlorophyll meter (Konica Minolta, Tokyo, Japan).

Individual photosynthetic pigments (mg g^−1^ fresh weight) were also determined using the method of Moran and Porath [47], and the results were analyzed using the method of Wellburn [48]. Fifty milligrams of fresh plant sample was taken from the fourth leaf, dissolved in 5 mL N, N-dimethylformamide at 4 °C for 72 h. The absorbance of the extracts was measured spectrophotometrically at 480, 647, and 664 nm wavelengths with a Nicolet Evolution 300 UV-Vis Spectrometer (Thermo Fisher Scienctific, Waltham, Massachusetts, USA). The quantity of photosynthetic pigments was measured from the fifth leaf at 21 d.a.s.

The contents of photosynthetic pigments were determined using the following formulas:chlorophyll-a = (11.65 × a664−2.69 × b647)chlorophyll-b = (20.81 × b647−4.53 × a664)carotenoids = (1000 × car480−1.28 × a664−56.7 × b647)total chlorophyll content = chlorophyll-a + chlorophyll-bchlorophyll a/b = chlorophyll-a/chlorophyll-b

### 4.3. Morphological Parameters

The stem diameter was measured with a slide caliper on the stem part between the second and third nodes. The plant height was scaled from the peat surface to the origin of the youngest leaf. These parameters were measured at 14 and 21 d.a.s. The root parameters (total length, root volume, diameter, tips, folks, surface, root diameters fractions) were measured after washing the roots. Roots were scanned with an Epson Expression 11,000 XL scanner and digitalized at a resolution of 400 dpi and analyzed using WinRHIZO Software (Regent Instrument Inc., Québec, QC, Canada). To determine the dry weight, different plant organs (shoot; root; total dry biomass) were collected and dried at 65 °C for three days, then the samples were measured by Ohaus AX223 analytical balance (Aqua-Terra Lab., Veszprém, Veszprém county, Hungary). These parameters were measured at 21 d.a.s.

### 4.4. Determination of Lipid Peroxidation

The method for determination of lipid peroxidation (from fresh leaf and root samples) was adopted from the methodology provided by Heath and Packer [49]. Malondialdehyde (MDA) content was determined at 21 d.a.s. from the fifth leaf. Root or leaf (0.1 g) samples were ground with liquid nitrogen and homogenized in 1 mL solution containing 0.25% (*w/v*) thiobarbituric acid (TBA) and 10% (*w/v*) trichloroacetic acid (TCA). Samples were centrifuged at 10,800× *g* for 25 min at 4 °C. The supernatant (0.2 mL) was transferred to a clean Eppendorf tube containing 0.8 mL solution of 0.5% (*w/v*) TBA and 20% (*w/v*) TCA. The mixture was heated at 95 °C for 30 min using a thermoshaker (Bioshan TS-100) and then cooled rapidly on ice. The absorbance was measured spectrophotometrically (Nicolet Evolution 300 UV-Vis Spectrometer) at 532 and 600 nm. The MDA content was calculated using the extinction coefficient of 155 mM^−1^ cm^−1^.

### 4.5. Antioxidant Enzyme Assays

Enzyme extract for the ascorbate peroxidase (APX), guaiacol peroxidase (POD), and protein content was prepared from the fifth leaf and root samples according to the method of Pukacka and Ratahczak [50]. Plant samples (0.2 g ground leaf or root) were homogenized to a fine paste in 1 mL 50 mM potassium phosphate buffer (pH 7) containing 2% (*w/v*) PVPP (polyvinylpyrrolidone), 1 mM ascorbate, 0.1% (*v/v*) Triton X-100, and 1 mM ethylenediaminetetraacetic acid (EDTA). Samples were centrifuged at 15,000× *g* for 20 min at 4 °C and the obtained supernatant was transferred into a clean Eppendorf tube and stored on ice until processing.

The method described by Mishra et al. [51] was adopted with some modifications to conduct the APX assay. The assay mixture (1 mL) contained 570 µL of 50 mM potassium phosphate buffer (pH 7.0), 200 µL H_2_O_2_ (0.1 mM), 150 µL sodium ascorbate (0.5 mM), 50 µL EDTA (0.1 mM EDTA), and 30 µL enzyme extract. A resulting decrease in absorbance due to ascorbate oxidation was measured at 290 nm for 5 min at 20 °C against a blank that contained phosphate buffer in place of the enzyme extract. The extinction coefficient of 2.8 mM^−1^ cm^−1^ was used.

The modified method of Zieslin and Ben-Zaken [52] was used to measure the POD activity. The mixture had 50 µL 0.2 M H_2_O_2_, 100 µL 50 mM guaiacol, 340 µL distilled water, 490 µL 80 mM phosphate buffer (pH 5.5), and 20 µL enzyme. The POD activity was determined based on the concentration of generated tetraguaiacol. The absorbance of the reaction compound was read at 470 nm for 3 min at 30 °C. The blank contained everything except the enzyme, which was replaced by 50 mM phosphate buffer. To estimate the concentration of tetraguaiacol formed, the extinction coefficient of 26.6 mM ^−1^ cm^−1^ was used.

To measure superoxide dismutase (SOD) activity, inhibition of photochemical reduction of NBT was followed up according to the method described by Giannopolities and Ries [53] and Beyer and Fridovich [54]. The 0.4 g powdered leaf was homogenized in 4 mL and 0.2 g powdered root sample was homogenized in 2 mL 50 mM phosphate buffer (pH 7.8) containing 0.1 mM EDTA, 1% (*w/v*) PVP, and 1 mM phenylmethanesulfonyl fluoride (PMSF). Samples were centrifuged at 10,000× *g* for 15 min at 4 °C. One unit of SOD is defined as the amount of enzyme required to cause 50% inhibition of the reduction of NBT as monitored at 560 nm. The measurement of protein content for the enzyme extract was determined using the method of Bradford [55].

The antioxidant enzyme activities and protein content were measured from the fifth leaf at 21 d.a.s.

### 4.6. Statistical Analysis

IBM SPSS Statistics 25 software was used for statistical analysis. Data normality was tested by Kolmogorov–Smirnov and Shapiro–Wilk tests, then the means were compared using an independent t-test at *p* < 0.05 [56]. Significant differences are marked by letters (a, b, c, d) in the manuscript.

## Figures and Tables

**Figure 1 plants-10-00982-f001:**
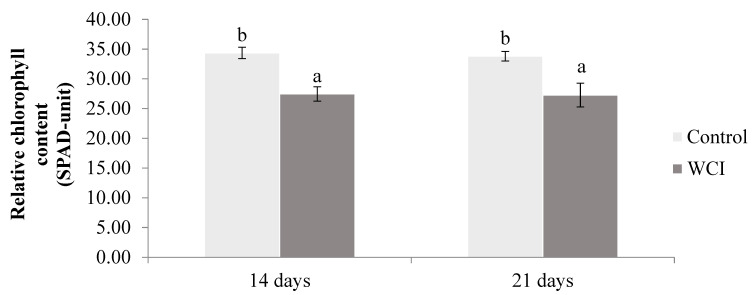
The effect of woolly cupgrass infestation (WCI) on the leaf relative-chlorophyll content of the fourth (14 d.a.s.) and fifth (21 d.a.s.) maize leaves (values represent means ± SD, *n* = 50). Small letters (a, b) show significant differences between treatments based on independent *t*-test (*p* < 0.05).

**Figure 2 plants-10-00982-f002:**
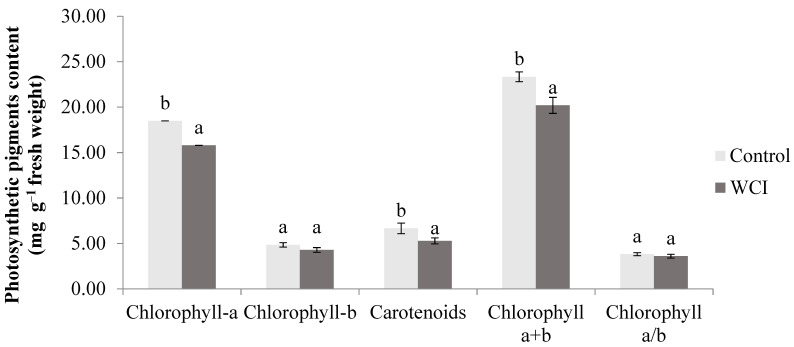
The effects of woolly cupgrass infestation (WCI) on the individual leaf photosynthetic pigments (chlorophyll-a, chlorophyll-b, and carotenoids) of the fourth (14 d.a.s.) maize leaf (values represent means ± SD, *n* = 10). Small letters (a, b) show significant differences between treatments based on independent t-test (*p* < 0.05).

**Figure 3 plants-10-00982-f003:**
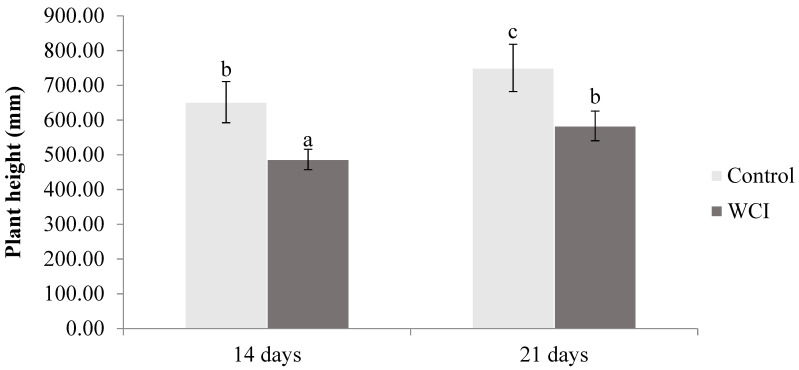
The effect of woolly cupgrass infestation (WCI) on the maize plant height (14 and 21 d.a.s.) (values represent means ± SD, *n* = 10). Small letters (a, b, c) show significant differences between treatments based on independent *t*-test (*p* < 0.05).

**Figure 4 plants-10-00982-f004:**
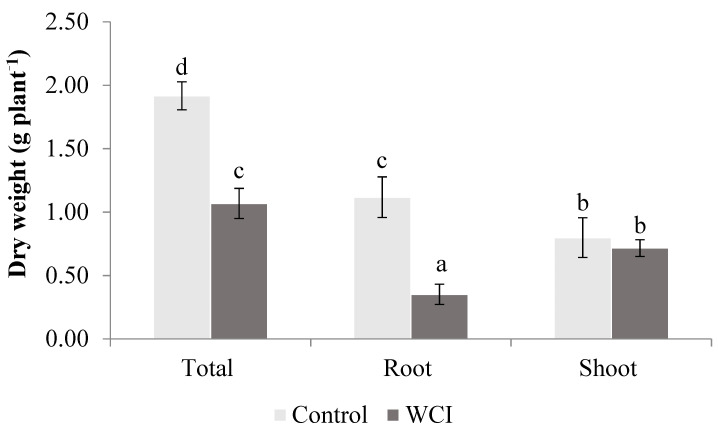
The effect of woolly cupgrass infestation (WCI) on the dry weights of maize 21 d.a.s. (values represent means ± SD, *n* = 5). Small letters (a, b, c, d) show significant differences between treatments based on independent *t*-test (*p* < 0.05).

**Figure 5 plants-10-00982-f005:**
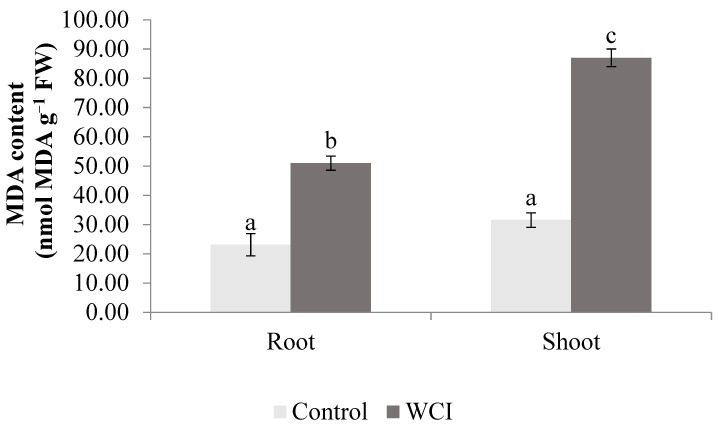
The effect of woolly cupgrass infestation (WCI) on the MDA content of maize 21 d.a.s. (values represent means ± SD, *n* = 5). Small letters (a, b, c) show significant differences between treatments based on independent *t*-test (*p* < 0.05).

**Figure 6 plants-10-00982-f006:**
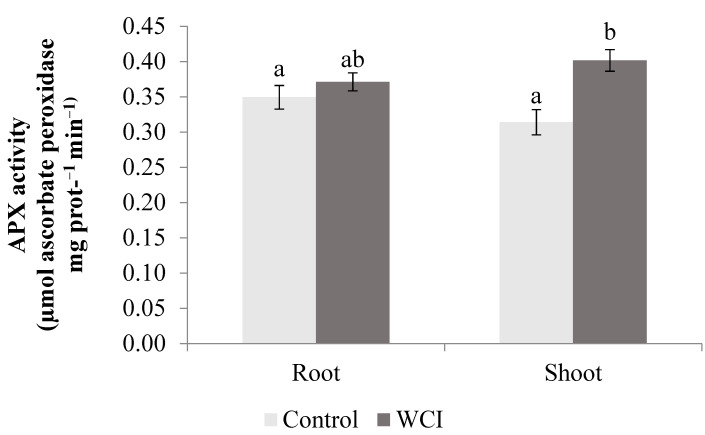
The effect of woolly cupgrass infestation (WCI) on the APX activity of maize 21 d.a.s. (values are means ± SD, *n* = 5). Small letters (a, b) show significant differences between treatments based on independent *t*-test (*p* < 0.05).

**Figure 7 plants-10-00982-f007:**
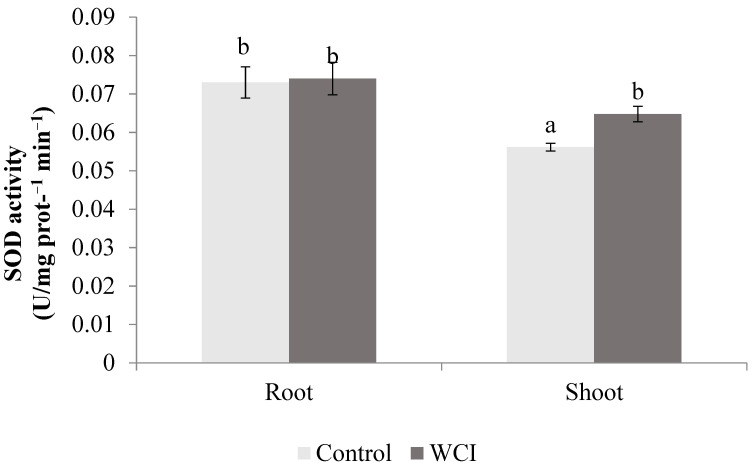
The effect of woolly cupgrass infestation (WCI) on the SOD activity of maize 21 d.a.s. (values are means ± SD, *n* = 5). Small letters (a, b) show significant differences between treatments based on independent *t*-test (*p* < 0.05).

**Figure 8 plants-10-00982-f008:**
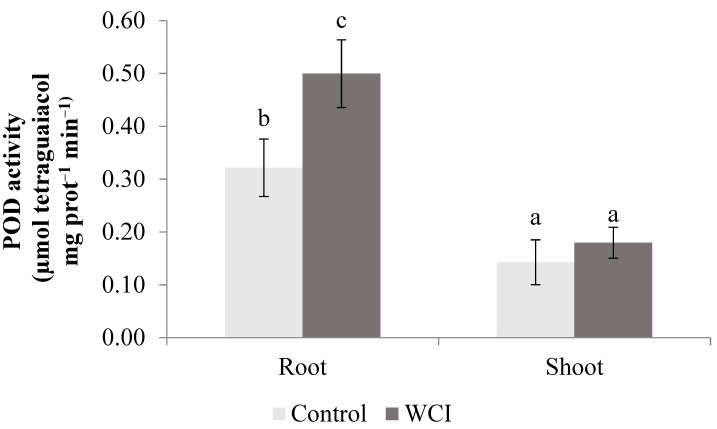
The effect of woolly cupgrass infestation (WCI) on the POD activity of maize 21 d.a.s. (values represent means ± SD, *n* = 5). Small letters (a, b, c) show significant differences between treatments based on independent *t*-test (*p* < 0.05).

**Figure 9 plants-10-00982-f009:**
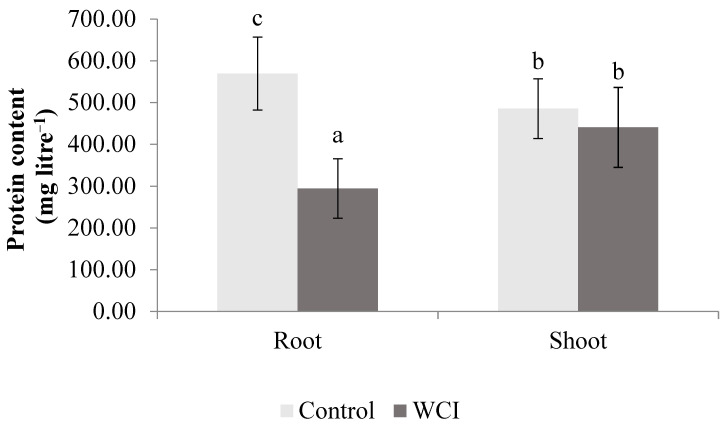
The effect of woolly cupgrass infestation (WCI) on the protein content of maize 21 d.a.s. (values represent means ± SD, *n* = 5). Small letters (a, b, c) show significant differences between treatments based on independent *t*-test (*p* < 0.05).

**Table 1 plants-10-00982-t001:** The effect of woolly cupgrass on the total root length (cm), root volume (cm^3^), average root diameter (mm), root surface (cm^2^), and root length based on root diameter fractions (cm) 21 d.a.s. (*n* = 5).

	Treatments	
Root Parameters	Control	WCI
Total length	757.17 ± 41.22 b	478.49 ± 79.62 a
Volume	1.54 ± 0.07 b	0.76 ± 0.11 a
Diameter	0.49 ± 0.01 a	0.46 ± 0.03 a
Surface	117.58 ± 12.05 b	67.24 ± 13.39 a
<0.2 mm	382.67 ± 30.58 b	283.89 ± 55.53 a
0.2–0.4 mm	118.95 ± 20.75 b	53.77 ± 8.15 a
0.4–0.6 mm	27.91 ± 4.85 b	11.21 ± 2.61 a
0.6–0.8 mm	61.81 ± 1.19 b	42.19 ± 3.45 a
0.8–1 mm	84.18 ± 6.18 b	41.10 ± 5.10 a
>1.0 mm	94.79 ± 7.48 b	46.23 ± 10.14 a

Small letters (a, b) show significant differences between treatment based on the independent *t*-test (*p* < 0.05).

## Data Availability

The data presented in this study are available upon request from the corresponding author.

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
