# Peer review of "The Impacts of Woolly Cupgrass on the Antioxidative System and Growth of a Maize Hybrid"

_plants, 2021, doi:10.3390/plants10050982_

Round 1
Reviewer 1 Report
Dear Author,
In this study methods are not clear,
1. In methods in was mentioned as 14th and 21st day. however in result it is denoted as first and second sampling on 7th and 14th days.
2. No details about pigments, lipid perodixation, and antioxidant activity information
3. Kindly give more explaination about how root volume and Surface was measured ?
With the existing result, it is not understandable about the wolly cupgrass interaction with maize
All the best !
Author Response
Dear Reviewer,
The authors wish to thank for the valuable comments of the reviewer and have implemented the suggested changes in the manuscript according to the reviewers’ comments. We hope that all the points have been adequately addressed. The changes that have been introduced in the revised manuscript are highlighted in red font, and the answers to the comments are given below.

Reviewer 2 Report
The manuscript “The impacts of woolly cupgrass on antioxidative system and phenotypic trait of Hungarian maize hybrid” examines the allelopathic effects of an invasive weed (Eriochloa villosa) on several morphological and biochemical traits of maize. The authors concluded that woolly cupgrass significantly inhibited the growth of maize.
This manuscript addresses an interesting topic with high relevance for maize production. In general, the manuscript contains valuable information about the allelopathic effects of woolly cupgrass and could be considerably improved through streamlining of the text to highlight the most important findings of the study. I am not convinced that the authors properly discussed their findings relevant to the angle they chose to present it, i.e, no appropriate inference about what these findings mean for the future production of maize.
Language is quite fluent, but some sentences include awkward expressions. Many grammatical or typographical errors also occur. Thus, it would be necessary to check the language throughout.
Given the problems described above I recommend that the manuscript should undergo substantial revision before it may become acceptable for publication in Plants. I also provide some broad areas of concern, and detailed comments below:
Specific comments:
Line 2: In my opinion, this title does not reflect the main objective of the study. Why do you use “phenotypic trait”? It will be more logical to use “growth”, for example. Why do you use the term “Hungarian”? Is Armagnac a typical cv. in Hungary?
Line 14: Replace “/Thunb./Kunth” by “(Thunb.) Kunth”
Line 15: Why do you use the term “physiology”? I do not consider MDA, leaf photosynthetic pigments and proteins as physiological parameters, but biochemical determinations.
Line 16: Please specify the maize cultivar.
Line 17-19: Please reformulate the sentence in order to clarify the methodology used. Antioxidant enzymes, MDA and proteins were measured in roots and shoots while photosynthetic pigments were determined in the leaves.
Line 27-28: What do you mean with “their role needs to be further explored, with reference to chlorophyll-b and shoot dry weight, which remained stable”? As a conclusion, at the end of the abstract, the authors should expose the relevance of their findings.
Line 31: Here, you use the term “lipid peroxidation” but it was not clear to which parameters you refer. When you mention “photosynthetic pigments” you should include “leaf”.
Line 40: The Introduction should be made more concise and should provide better background information about the effects of weeds on maize yield. It's true there is little information on the weed Eriochloa villosa. However, this topic has been covered widely in other ones. I think it could be included a more extended paragraph describing in more detail how such weeds affected the crop yield. The authors only presented one paragraph about it and this seems not enough.
Line 41: Please consider changing “functioning” by “function”.
Line 44: “agricultural” seems to be redundant.
Line 76: Why do mention several ROS species if you did not include their determination in the study? In my opinion, the authors should focus on the same points that they determined, i.e., growth, antioxidant enzymes, lipid peroxidation, proteins and leaf photosynthetic pigments.
Line 87: Why this mention to “yield”? What do you mean when you say " this research show how can correlate between the effect of weed infestation and the changes of plant physiological parameters ". You only use correlations between antioxidant enzymes and proteins.
Line 92-93: Please clarify the dates/time of observations: 1 and 14 days or 14 and 21 days as expressed in lines 263?
Line 107: Why do you use the term “interestingly”?
Line 110: I cannot understand the use of “smaller”.
Line 118-119: Please reformulate the sentence and replace “allelophatic” by “allelopathic”.
Line 146-148: As SOD presented the same pattern as APX, you can reformulate all the sentence (see lines 140-142).
Line 166: Correlation between variables can be or minor relevance, and even misleading for readers. Why do you only perform correlation between antioxidant enzymes and proteins?
Line 173: The Discussion of the results is too general. The authors should add more discussion relating the major findings of the manuscript and should on the implications of study results.
Line 174: Why do you use the term “interaction”?
Line 177-178. This sentence is more appropriate for the Introduction.
Line 179-180: This sentence is not true because there was no variation in chlorophyll-b. Please Rewrite the sentence in accordance with the statistical analysis (post-hoc).
Line 171: Please remove “Figure 2”.
Line 183-186: How do you explain the lack of significance in chlorophyll-b?
Line 193: This argument“…which corresponds to the decline in the chloroplast pigments” is far-fetched without a clear support by the results. I think that the lowest performance under WCI could be related to poor photosynthetic capacity of maize but this was not determined in the present study.
Line 217-220: “In agreement, Hatata and El-Darier [37] found that the 217 activity of APX was higher when the wheat was treated by Achillea santolina shoot extract. Gindri et al. [38] also stated that Lantana camara L. increased the APX activity of the host plant (Avena sativa L.)”. It was not clear if higher APX activity was found in shoots or roots. Please clarify it.
Line 220-221. The sentence “According to Ding et al. [39], higher SOD activity was measured in the garlic (Allium sativum L.) root exudates treated lettuce extract” contracted the results of the current study. Please revise it.
Line 224-227: It was not clear if higher POD activity was found in shoots or roots. Please clarify it.
Line 233-235: The results did not support this argument. Please revise the text.
Line 249: It could be interesting to include one sentence about the major implications of this study to the management of maize stands invaded by woolly cupgrass.
Line 251: In the Material and Methods, the presentation of the methodology is not sufficient. For example, the authors did not explain the number of replicas per parameter, dimension of pots, and did not define the sampling time of the majority of determinations.
Other comments:
Figures: Presentation of all figures should be revised in order to be more informative and appellative. For example, the order of treatments (Control and WCI) should be the same in all figures; all horizontal lines should be removed. Moreover, the y-axis should be reformulated. In the case of Figure 1, the title should be “Relative chlorophyll content (SPAD-units)” and the first sampling date and second sampling time should be replaced by the day. The same occurred in Figure 3. In Figure 2 to Figure 9, it will be important to indicate the sampling date. In Figure 4, in the x-axis should appear “total”, root”, and “shoot”. In Figure 5 to Figure 8, the y-axis should be replaced by each abbreviature.
Tables 2: It should be included a note for “NO” that I suppose means “non significant correlation”.
References: Please make sure that all references meet the assumptions of the journal. For example, the way in which some journal title is cited is incorrect. Please verify all of them.
Author Response

(The authors gave the same response as above.)

Round 2
Reviewer 1 Report
Author has answered all my questions properly.
Thank you!
All the best.
Author Response
Dear reviewer I.,
Thanks for your suggestions, reviewing our manuscript.
The authors wish all the best for you!
Reviewer 2 Report
I´m glad to see authors have improved the first version of the manuscript. The major concerns made in the first review have been corrected and now I suggest that the manuscript can be considered for publication.
Only some remarks:
Line 90: Replace “21 days after sowing” by “21 d.a.s.”
Fig. 1 and Fig. 3: Replace “14 days after sowing” by “14 d.a.s.” and “21 days after sowing” by “21 d.a.s.”
Table 1: Use the same decimal places in all root parameters. Please verify the volume and diameter under the Control treatment
In the Material and Methods section, revise the use of d.a.s. in lines 267, 271, 272, 280, 291, 298, 302 and 344
Author Response
Dear reviewer II.,
Thanks for your suggestions, reviewing our manuscript.
The authors wish all the best for you!
